# Inhibition of miR-152 during In Vitro Maturation Enhances the Developmental Potential of Porcine Embryos

**DOI:** 10.3390/ani10122289

**Published:** 2020-12-04

**Authors:** Ahmed Gad, Matej Murin, Lucie Nemcova, Alexandra Bartkova, Jozef Laurincik, Radek Procházka

**Affiliations:** 1Laboratory of Developmental Biology, Institute of Animal Physiology and Genetics of the Czech Academy of Sciences, 27721 Libechov, Czech Republic; Murin@iapg.cas.cz (M.M.); Nemcova@iapg.cas.cz (L.N.); alexandra.bartkova@ukf.sk (A.B.); laurincik@gmail.com (J.L.); Prochazka@iapg.cas.cz (R.P.); 2Department of Animal Production, Faculty of Agriculture, Cairo University, Giza 12613, Egypt; 3Faculty of Natural Sciences, Constantine the Philosopher University in Nitra, 94901 Nitra, Slovakia

**Keywords:** miR-152, porcine, oocyte, blastocyst rate

## Abstract

**Simple Summary:**

MiR-152 is a highly conserved miRNA across different species and plays a role in the regulation of cell differentiation, proliferation, and survival. However, the exact role of miR-152 in oocyte and embryo development is not yet known. In this study, we specifically manipulated the expression level of miR-152 in porcine cumulus-oocyte complexes (COCs) and monitored their developmental competence until the blastocyst stage. We mainly found that a suppressed expression of miR-152 during oocyte maturation significantly improved the blastocyst rate. Our results indicate that this negative correlation between miR-152 during oocyte maturation and the blastocyst rate in pigs could be through targeting IGF system components during oocyte development. These results provide more insights into the role of miRNAs during oocyte and embryonic development that could improve the in vitro production system for mammalian embryos.

**Abstract:**

Oocyte developmental competence is regulated by various mechanisms and molecules including microRNAs (miRNAs). However, the functions of many of these miRNAs in oocyte and embryo development are still unclear. In this study, we managed to manipulate the expression level of miR-152 during oocyte maturation to figure out its potential role in determining the developmental competence of porcine oocytes. The inhibition (Inh) of miR-152 during oocyte maturation does not affect the MII and cleavage rates, however it significantly enhances the blastocyst rate compared to the overexpression (OvExp) and control groups. Pathway analysis identified several signaling pathways (including PI3K/AKT, TGFβ, Hippo, FoxO, and Wnt signaling) that are enriched in the predicted target genes of miR-152. Gene expression analysis revealed that *IGF1* was significantly up-regulated in the Inh group and downregulated in the OvExp group of oocytes. Moreover, *IGF1R* was significantly upregulated in the Inh oocyte group compared to the control one and *IGFBP6* was downregulated in the Inh oocyte group compared to the other groups. Blastocysts developed from the OvExp oocytes exhibited an increase in miR-152 expression, dysregulation in some quality-related genes, and the lowest rate of blastocyst formation. In conclusion, our results demonstrate a negative correlation between miR-152 expression level and blastocyst rate in pigs. This correlation could be through targeting IGF system components during oocyte development.

## 1. Introduction

The posttranscriptional regulation of maternal RNA during oocyte development is one of the main mechanisms controlling oocyte developmental competence. Different factors and molecules are involved in this mechanism by interacting with specific RNA sequences [1,2]. MicroRNAs (miRNAs), short non-coding RNA molecules (~22 nucleotides long), are important actors that regulate gene expression posttranscriptionally through different mechanisms including translational inhibition or the degradation of complementary mRNA sequences [3]. In mammals, there is cumulative evidence that indicates the essential roles of miRNAs in various reproductive processes including gametogenesis, folliculogenesis, oocyte maturation, and embryonic development (for a review, see [4]). It has been suggested that miRNAs could regulate follicular and oocyte development mainly through regulating gene expression in the ovarian somatic cells, such as cumulus and granulosa cells [5,6]. It is well known that the paracrine signaling and intercellular communication between oocyte and cumulus cells (CCs) are key factors affecting oocyte developmental competence [7]. This bidirectional communication was reported to be an essential regulator of miRNA expression in both cell types [8]. Moreover, there is also evidence for a direct role of maternal miRNAs in regulating oocyte mRNAs and controlling the maternal-zygotic transition [9,10]. Several studies investigated the function of specific miRNAs during oocyte and embryonic development by modulating their expression levels after microinjection or transfection with the corresponding precursors and inhibitors. For instance, the injection of pig oocytes with miR-27a mimics improved the first cleavage rate, which significantly decreased after injection with a miR-27a inhibitor [11]. Moreover, the overexpression of miR-378 in cumulus-oocyte complexes (COCs) suppresses cumulus expansion and reduces the developmental competence of porcine oocytes [12]. Modulating the expression level of miR-29b in mouse zygotes led to DNA methylation disorders and significantly blocked embryonic development from the morula to the blastocyst stage [13]. All these studies and others shed light on the regulatory roles of some miRNAs during oocyte and embryonic development. However, more research is still needed to discover the role and mechanism of action of other miRNAs to improve our understanding of oocyte developmental competence.

MiR-152, a member of the miR-148/152 family, is a highly conserved miRNA across different species and plays a role in the regulation of cell differentiation, proliferation, and survival [14]. In cancer research, miR-152 is well known as an inhibitor of cancer cell proliferation through suppressing the PI3K/Akt and MAPK signaling pathways [15]. In pigs, the miR-152 gene (miRBase Tracker: MI0013104) is located on the reverse strand of chromosome 12 within the intronic region of the COPI coat complex subunit zeta 2 gene (*COPZ2*). Recently, we reported that miR-152 was highly expressed in porcine oocytes aspirated from large compared to small ovarian follicles, as a model of oocyte competence [16]. Being a suppressor of cell proliferation and predictably targeting the oocyte meiosis pathway, we suggested that miR-152 could play a role in the maintenance of oocyte meiotic arrest, especially in large oocytes that are competent to resume meiosis [16]. However, to the best of our knowledge, the exact role of miR-152 in oocyte and embryo development is not yet known. Therefore, the objective of this study was to figure out the potential role of miR-152 in determining the developmental competence of porcine oocytes. We specifically manipulated the expression level of miR-152 in porcine COCs and monitored their developmental competence until the blastocyst stage. In addition, we analysed the expression level of different predicted miR-152 target genes in the oocytes and quality-related genes in the produced blastocysts.

## 2. Materials and Methods 

All chemicals were purchased from Sigma-Aldrich (Munich, Germany) unless otherwise stated. All plastic materials were purchased from Nunc (Roskilde, Denmark) unless otherwise stated.

### 2.1. Oocyte Collection

Porcine ovaries were collected from prepubertal gilts at a local abattoir and transported to our laboratory in a thermos flask within 2 h of their slaughter. Ovaries were washed three times with saline solution. Cumulus-oocyte complexes (COCs) were aspirated from large follicles (3–6 mm in diameter) using a 20-gauge needle attached to a 10 mL syringe in PXM-HEPES (HEPES buffered porcine X medium, [17]). The diameter of follicles was measured as previously recommended [18]. COCs were morphologically analyzed under a stereomicroscope (Zeiss Stemi 508, magnification × 50). Only those with at least three layers of CCs and an evenly granulated ooplasm were used for our experiments.

### 2.2. COCs Transfection and IVM

Chemically modified double-stranded RNAs that specifically mimic miR-152 and single-stranded RNAs that specifically bind to and inhibit miR-152 (ThermoFisher Scientific, Waltham, MA, USA) were used to overexpress (OvExp) and inhibit (Inh) the expression level of the endogenous miR-152 in porcine oocytes, respectively. Transfection of a scrambled miRNA sequence (ThermoFisher Scientific) was used as a negative control (Neg. control) and non-transfected oocytes were used as a control group. 

Lipofectamine 3000 (1.5%, ThermoFisher Scientific) was used to transfect COCs with 40 nM of miR-152 mimic or negative control or with 600 nM of miR-152 inhibitor. Each miRNA treatment was mixed with Lipofectamine 3000 in Opti-MEM Medium I (ThermoFisher Scientific) and incubated at room temperature for 30 min. A total of 50 µL of the transfection solution was added to 550 µL of the maturation medium. 

COCs were washed twice in Medium 199 supplemented with 0.005% gentamicin (Roth 0233), 0.0022% sodium pyruvate, 0.01% L-glutamine and 0.1% BSA. Around 50 COCs were cultivated in 4-well dishes for 4 h in 600 µL of the medium supplemented with 1% dbcAMP (Dibutyryladenosine 3′, 5′-cyclic monophosphate sodium salt) and each of the transfection solutions at 38.5 °C under a 5% CO_2_ atmosphere. After that, transfection was continued in 4-well dishes for 44 h in 600 µL of our modified Medium 199 supplemented with 10 ng/mL EGF, 40 ng/mL FGF2, 20 ng/mL IGF1, 2000 IU/mL LIF (Leukemia Inhibitory Factor human, Merck, Prague, Czech Republic), 0.57 mM L-Cysteine, 10 IU/mL PMSG and 10 IU/mL HCG at 38.5 °C under a 5% CO_2_ atmosphere. The control group was cultivated in 600 µL of maturation medium without any transfection solution. After cultivation, CCs were removed from COCs as described below. A group of oocytes was mounted on glass slides and fixed in acetic alcohol for 48 h. Then, oocytes were stained with 1% orcein and evaluated with a light microscope with phase contrast and scored for the metaphase II (MII) stage. Another group of oocytes was used for parthenogenetic activation. Lastly, the third group of oocytes was stored in a lysis buffer at −80 °C until further analysis.

### 2.3. Parthenogenetic Activation and Cultivation of Embryos

Parthenogenetic development rather than in vitro fertilization (IVF) is now often used for the assessment of cytoplasmic maturation of cultured pig oocytes. This approach is justified by the fact that mammalian parthenogenetic or gynogenetic embryos undergo normal preimplantation development [19], and avoids the problem with polyspermy associated with pig IVF in vitro. For that, cumulus cells were removed from COCs by pipetting and washed twice in PXM-HEPES. Oocytes were activated by exposure to 10 µM ionomycin in PXM-HEPES for 5 min. After that, they were washed twice in porcine zygote medium 3 (PZM 3) [20] supplemented with 2 mM 6-DMAP and cultivated for 5 h at 38.5 °C under a 5% CO_2_ atmosphere. Around 50 putative parthenotes were washed twice in PZM 3 and cultivated for 6 days in 4-well dishes in 1 mL of PZM 3 medium at 38.5 °C under a 5% CO_2_ atmosphere. After 40 h, the cleavage of embryos was assessed and after 144 h, the ability of embryos to reach the blastocyst stage was analyzed.

### 2.4. Expression Analysis of miR-152 by Droplet Digital PCR (ddPCR)

Three biological replicates of pooled oocytes (*n* = 240 in total) and blastocysts (*n* = 120 in total) from control and treatment groups were used for the expression analysis. Total RNA enriched with miRNA was isolated from denuded oocytes and blastocysts of the different groups using a *mir*Vana miRNA Isolation Kit (Life Technologies, Carlsbad, CA, USA) according to the manufacturer’s instructions. Total RNA quality and integrity were assessed with an Agilent 2100 Bioanalyzer using an Agilent RNA 6000 pico kit (Agilent Technologies, Santa Clara, CA, USA). The abundance of miR-152 was quantified using a TaqMan miRNA assay (Applied Biosystems, Foster City, CA, USA) according to the manufacturer’s instructions. Briefly, 10 ng total RNA were reverse transcribed using a TaqMan MicroRNA Reverse Transcription Kit (Thermo Fisher Scientific, Waltham, MA, USA). RT reaction mixtures (15 μL) were prepared with 0.15 μL dNTPs (100 mM), 1.5 μL 10 X RT buffer, 0.19 μL RNase inhibitor (20 IU/μL), 1 μL reverse transcriptase (50 IU/μL), 3 μL of miRNA-specific stem-loop primer, and 5 μL input RNA. The reaction mixture was incubated at 16 °C, then 42 °C for 30 min., followed by 85 °C for 5 min. A QuantaLife QX200 ddPCR system (Bio-Rad Inc., Hercules, VA, USA) was used for miRNA quantification as previously described [16]. The expression of miR-152 was normalized to miR-125b and miR-26a, as both exhibited high stability in porcine oocytes [16]. In addition, miR-26a has been previously reported to be the most stable expressed miRNA in porcine oocytes [21]. 

### 2.5. Target Gene Prediction and Ontological Classification (In Silico Analysis)

MiR-152 predicted target genes were identified using the miRecords web-based database [22]. Genes predicted by at least four different algorithms were submitted to the DAVID Bioinformatics web-tool for pathway analysis [23]. Significant pathways were characterized with the Kyoto Encyclopaedia of Genes and Genomes (KEGG) database [24]. Interactions between targeted genes as well as the identified relevant pathways were visualized with the software Cytoscape [25] and the application ClueGO [26]. 

### 2.6. Expression Analysis of Predicted Target and Quality-Related Genes Using RT-qPCR

A group of predicted target genes related to oocyte maturation and development was selected and analysed in the different oocyte groups. In addition, blastocyst quality-related genes were analysed in the different blastocyst groups. Primers for the selected target and quality-related genes were designed using the software Beacon Designer v. 8.21 and listed in Appendix A. The one-step RT-qPCR was conducted in a RotorGene 3000 cycler (Corbett Research, Mortlake, Austria) using the QIAGEN OneStep RT-PCR Kit (Qiagen, Germany) in a 20 µL reaction mixture containing 4 µL 5 X reaction buffer, 0.8 µL dNTP mix (10 nM stock), 0.4 µL forward and reverse primers (20 nM stock), 0.125 µL RNasine (20 IU/mL stock, Promega), 0.8 µL enzyme mix, 0.8 µL EvaGreen (Biotium, CA, USA), 3 µL RNA, and nuclease-free water. The reaction conditions were as follows: reverse transcription at 50 °C for 30 min, initial denaturation at 95 °C for 15 min, followed by PCR cycles consisting of denaturation at 94 °C for 15 s, annealing at a temperature specific for each set of primers (Appendix A) for 15 s and extension at 72 °C for 20 s; and a final extension at 72 °C for 5 min. Fluorescence data were acquired at the end of each extension step. Products were verified by melting analysis and gel electrophoresis on 1.5% agarose gel with MidoriGreen Direct (Nippon Genetics, Dueren, Germany). Comparative analysis software (Corbett Research) was used for gene expression analyses after normalization to the geometric mean of *H1FOO* and *TUBA1B* or *H2AFZ* and *RPL19* mRNA abundance as internal control genes for oocytes and blastocysts, respectively. 

### 2.7. Statistical Analysis

At least three replicates were done for each treatment. Data are expressed as means ± SEM. Statistical analysis of mRNA expression was performed using a one-way ANOVA followed by multiple pairwise comparisons using the Tukey test. Meiotic competence and the development of embryos were analyzed by one-way ANOVA with the Holm–Sidak post-test (SigmaPlot 12.0, London, UK). Probability values < 0.05 were considered to be statistically significant.

## 3. Results

### 3.1. Developmental Potential of Oocytes and Embryos

In total, 294 oocytes were used to determine their maturation ability. There were no significant differences between the different groups in maturation rate after 44 h of IVM (Table 1). In total, 338 oocytes were used to determine their developmental potential. There were no significant differences in the cleavage rate between groups. However, the miR-152 Inh group exhibited a significantly higher blastocyst rate (48.69 ± 2.72) than the OvExp and control groups (Table 1).

### 3.2. Expression Pattern of miR-152 in Oocytes and Blastocysts

Absolute quantification using ddPCR was performed to measure the expression level of miR-152 in matured oocytes and blastocysts after oocyte treatment with miR-152 mimic or inhibitor. Expression analysis found a significantly higher level of miR-152 in the oocyte and blastocyst OvExp groups than in the other groups (Figure 1). Inhibition treatment found a significant downregulation pattern of miR-152 in oocytes compared to the negative control and in blastocysts compared to the control group (Figure 1). 

### 3.3. Target Gene Prediction and Pathway Analysis

Genes that were predictably targeted by miR-152 were identified to understand the potential function of miR-152 during oocyte and embryo development. A total of 636 different transcripts were predicted to be targeted by miR-152 (Appendix A). Pathway analysis revealed that 16 canonical pathways were significantly enriched in the predicted target genes. Signaling pathways (including TGFβ, FoxO, regulating the pluripotency of stem cells, PI3K-Akt, Hippo, HIF-1, mTOR, and Wnt), cellular process pathways (including circadian rhythm, focal adhesion, regulation of actin cytoskeleton, cell cycle, and oocyte meiosis), and pathways in cancer were among the significant pathways enriched in the target genes (Appendix A). Interaction networking between the target genes and their corresponding highly enriched pathways are presented in Figure 2.

### 3.4. Gene Expression Patterns in Oocytes and Blastocysts

The RT-qPCR analysis was performed to measure the relative expression for a group of miR-152 predicted target genes in matured oocytes and quality-related genes in blastocysts. In the oocyte groups, two genes exhibited a clear opposite pattern of expression in OvExp and Inh groups compared to the controls. *IGF1* was significantly downregulated in the OvExp and upregulated in the Inh oocyte group, while *PPP1CB* exhibited the opposite pattern. Moreover, *IGF1R* and *IGFBP6* were up- and downregulated, respectively, in the Inh oocyte group compared to the control (Figure 3A). In blastocysts, the *BCL2* gene was upregulated in the OvExp group compared to the Inh and Neg. control groups, and the *BAX* gene was downregulated in the Inh compared to the control group. On the other hand, the *SOD1* gene was downregulated in the OvExp compared to other blastocyst groups, and the *GJA1* gene exhibited no significant differences between all blastocyst groups (Figure 3B). 

## 4. Discussion

In our previous study, we compared the miRNA expression profile of the growing and fully grown porcine oocytes based on the follicle size using small RNA high-throughput sequencing technology [16]. Results showed that miR-152 was among the significantly upregulated miRNAs in the large fully grown oocytes compared to the small growing ones. To verify the role of miR-152 in determining the developmental competence of porcine oocytes, in this study we managed to manipulate the expression level of miR-152 in porcine COCs during the maturation process. We mainly found that a suppressed expression of miR-152 during oocyte maturation significantly improved the blastocyst rate. Recently, it has been reported that the miRNAs expression pattern is highly correlated with oocyte quality and developmental competence [27]. Moreover, stage-specific and dynamic changes in the expression pattern of several miRNAs were observed during oocyte [28,29,30] and embryo development [8,31], indicating a regulatory role of miRNAs in oogenesis and embryogenesis. Functional analysis studies revealed the specific function of several miRNAs as post-transcriptional regulators during oocyte and embryo development. For instance in pigs, miR-224 has been reported to be a regulator of the CCs function and subsequently affect oocyte maturation [32]. Moreover, manipulating the expression level of miR-155 in mice COCs proved its role in CCs expansion, oocyte maturation, and cleavage rate [33]. Several other miRNAs, such as let-7c, miR-27a, and miR-322, have been investigated and showed a complex regulatory role of oocyte developmental competence through functional regulation of the surrounding follicular cells [34]. These evidences highlighted the substantial roles of miRNAs in mammalian reproduction throughout folliculogenesis, oogenesis, and embryo development (for a review, see [4]). 

It is well known that miR-152 plays an important role as a tumor suppressor in human cancer [14]. Several mechanisms have been demonstrated to explain this role, including the epigenetic regulation of miR-152/DNMT1 [35], the inhibition of cell proliferation [36], and suppression of the PI3K/Akt and MAPK signaling pathways [37]. In the current study, we successfully manipulated the expression level of miR-152 in porcine COCs via the transfection of miR-152 mimics and inhibitors. The quantitative analysis of PCR data showed a significant reduction and increase in the miR-152 expression level in Inh and OvExp oocyte groups, respectively. Although we confirmed this manipulation into the matured oocytes, we cannot neglect the same effect on the surrounding CCs that could indirectly affect oocyte competence. However, in this study, we focused on the miRNA and gene expression alterations in the matured oocytes and the subsequent effects on the produced blastocysts. In bovine, a stable expression pattern of miR-152 has been detected in GV, in vitro matured oocytes, and zygotes [38]. This may indicate that in vitro culture conditions do not affect the expression level of this particular miRNA. However, comparing the expression patterns under in vivo conditions will give more insights into the role of miR-152 in oocyte competence. Target gene prediction and pathway analysis revealed that miR-152 could be a potential candidate for controlling oocyte development through targeting different essential pathways and genes, including *IGF1* and its receptor *IGF1R*. Gene expression analysis of oocyte groups showed that *IGF1* was significantly upregulated in the Inh group and downregulated in the OvExp group, while *IGF1R* was significantly upregulated in the Inh oocyte group compared to the control one. This may indicate a potential regulatory mechanism between miR-152 and the IGF system during oocyte development. Several studies reported that IGF-1 is involved in the regulation of follicular, oocyte, and embryonic development and the proliferation of follicular cells [39,40,41]. A significant correlation between *IGF1* expression and blastocyst formation has been observed in mice [42]. In other mammalian species, including pigs, it has been confirmed that IGF-1 improves in vitro maturation, fertilization, and blastocyst rates [43,44,45,46,47]. In support of our results, the association between IGF-1 and miR-152 expression has been previously demonstrated in human oocytes, since the culturing of immature oocytes in the presence of IGF-1 significantly reduced the expression of miR-152 [48]. Moreover, in cancer research, it was suggested that miR-152 is involved in the IGF-1-mediated miR-152/PKM2/β-catenin regulatory circuit that regulates cell proliferation and angiogenesis [49]. Additionally, the overexpression of miR-152 inhibits breast cancer cell proliferation via targeting IGF1R and IRS1 and suppressing their downstream AKT and MAPK/ERK signaling pathways [37]. We also measured the expression level of two IGF-binding proteins (*IGFBP6* and *IGFBP7*) in the different oocyte groups. The results showed a significant reduction in *IGFBP6* in the Inh oocyte group compared to other groups. IGF-binding proteins control the bioavailability of IGFs to their corresponding cells [50]. The expression levels of IGFBP-2 mRNA and protein were reduced during oocyte maturation in bovines [51]. Previously we reported a significant decrease in *IGFBP2*, *6*, and *7* in the high- compared to low-competence porcine oocytes [52]. These results in addition to our current findings may indicate that the improved blastocyst rate after the inhibition of miR-152 in the COCs is due to the higher IGF bioavailability through the upregulation of *IGF1* and *IGF1R* and the reduction in *IGFBP6* during oocyte development.

Pathway analysis identified several signaling pathways (including PI3K/AKT, TGFβ, Hippo, FoxO, and Wnt signaling) that are enriched in predicted miR-152 target genes. The expression of these signaling pathway components is essential for the oocyte to embryo transition and during the early implantation stage, including blastocyst formation [53,54]. For instance, the inhibition of PI3K or AKT, components of the PI3K/AKT signaling pathway, eventually produces a significant decrease in embryo development and blastocyst rate in bovines [55,56]. In this study, blastocysts developed from the mimic-treated oocytes exhibited an increase in miR-152 expression compared to the other groups. This blastocyst group represents the lowest rate of blastocyst formation, which indicates a negative effect of miR-152 on embryonic development. In agreement with our results, Nie et al. [57] reported that the transfection of specific miR-152 mimics into the endometrial epithelial cells led to impaired embryonic development and implantation. The same blastocyst group (OvExp) exhibited a reduction in the *SOD1* expression pattern, suggesting a lower antioxidant capacity of these blastocysts. Recently, it has been reported that miR-152 expression is positively correlated with oxidative stress in HL-2 cells, in which a dose-dependent high expression of miR-152 was observed after treatment with hydrogen peroxide (H2O2) [58]. Although the oxidative stress and the reduction in SOD levels could lead to cell apoptosis in preimplantation embryos [59], the expression of the antiapoptotic gene *BCL2* was significantly upregulated in the blastocysts of the OvExp group compared to the Inh and Neg. control blastocyst groups. In agreement with our results, Cao et al. reported that the transfection of miR-152 mimics significantly increases *BCL2* expression and inhibits apoptosis induced by hypoxia in HBMECs cells [60]. 

## 5. Conclusions

Our results demonstrate a negative correlation between miR-152 expression level and blastocyst rate in pigs. The overexpression and inhibition of miR-152 during oocyte maturation dysregulate the expression level of *IGF1*, *IGF1R*, and *IGFBP6* in the treated oocytes and *SOD1* and *BCL2* in the produced blastocysts. Further investigations of the exact mechanisms by which miR-152 and other miRNAs regulating oocyte and embryo development in comparison with in vivo produced ones are necessary to improve our knowledge and subsequently improve the in vitro production system for mammalian embryos.

## Figures and Tables

**Figure 1 animals-10-02289-f001:**
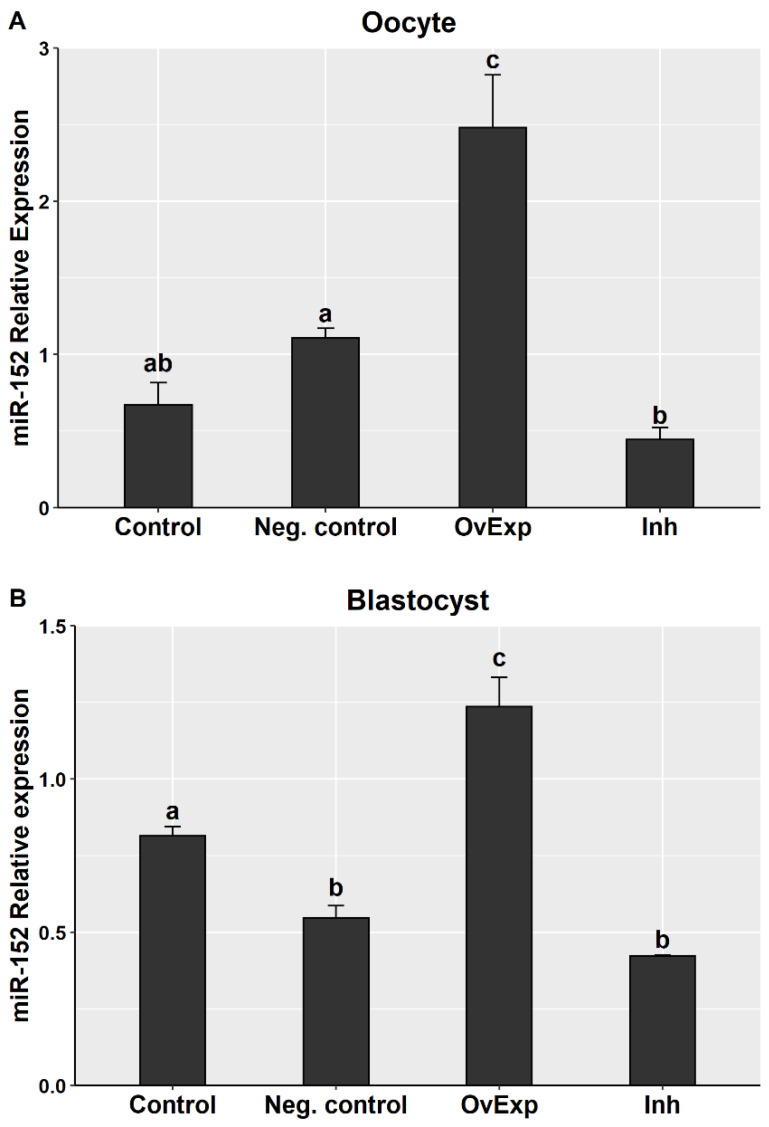
Relative expression of miR-152 in matured oocytes (**A**) and blastocysts (**B**). OvExp: miR-152 overexpression; Inh: miR-152 inhibition; Neg. control: negative control. Different letters indicate significant differences between groups (*p* < 0.05).

**Figure 2 animals-10-02289-f002:**
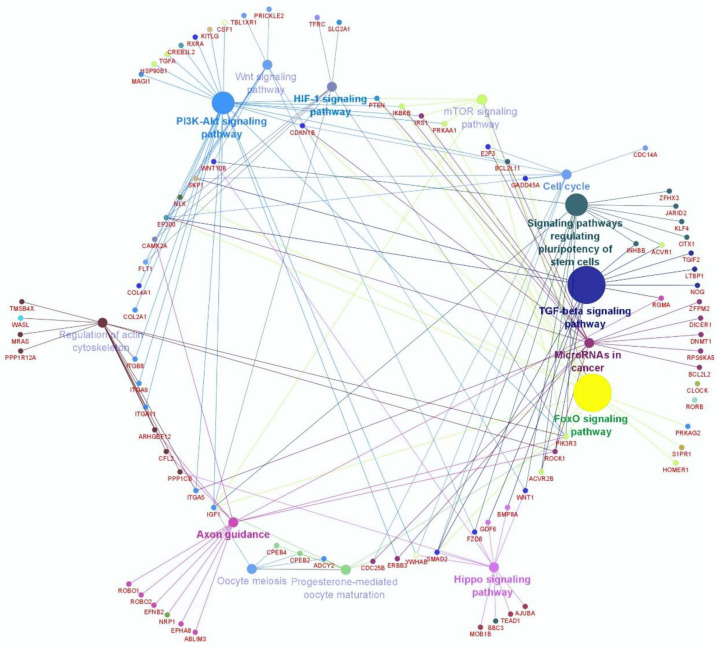
Interaction networking of predicted miR-152 target genes and their corresponding highly enriched pathways.

**Figure 3 animals-10-02289-f003:**
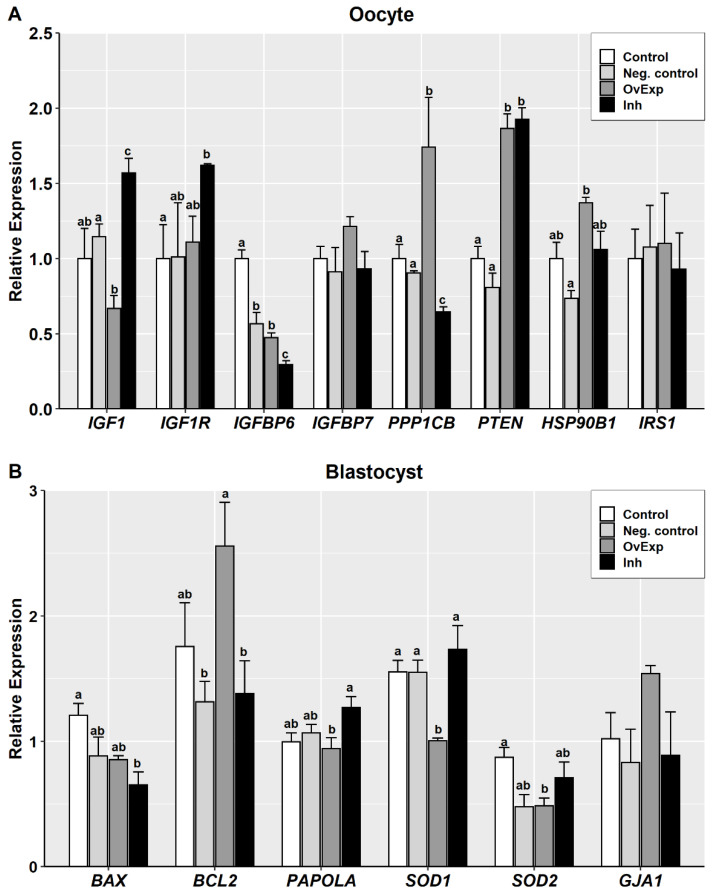
Expression of predicted miR-152 target genes in matured oocytes (**A**) and quality-related genes in blastocysts (**B**). OvExp: miR-152 overexpression; Inh: miR-152 inhibition; Neg. control: negative control. Different letters indicate significant differences between groups (*p* < 0.05).

**Table 1 animals-10-02289-t001:** Maturation rate of porcine oocytes and the developmental potential of embryos after parthenogenetic activation.

Group	Metaphase II Rate (% ± SEM)	Cleavage Rate (% ± SEM)	Blastocyst Rate (% ± SEM)
Control	92.49 ± 6.29	82.15 ± 9.24	28.12 ± 2.70 ^a^
Neg. control	92.22 ± 1.57	70.82 ± 12.33	28.38 ± 4.69 ^a^
OvExp	91.87 ± 3.40	76.95 ± 2.49	22.19 ± 8.92 ^A,a^
Inh	90.29 ± 6.94	82.08 ± 5.25	48.69 ± 2.72 ^B,b^

OvExp: miR-152 overexpression; Inh: miR-152 inhibition; Neg. control: Negative control. ^A,B^ significantly different within column (*p* < 0.01), ^a,b^ significantly different within column (*p* < 0.05).

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
