# Peer review of "Inhibition of miR-152 during In Vitro Maturation Enhances the Developmental Potential of Porcine Embryos"

_animals, 2020, doi:10.3390/ani10122289_

Round 1
Reviewer 1 Report
The aim of this study is to investigate the influence of miR-152 on oocyte developmental competence and the pathways involved.
The paper is written in a comprehensive and concise way. The introduction provides an informative background leading to a very clearly described aims. The methods, results and discussion are also very clear and easy to follow. The conclusion is strongly substantiated by the described observations.
I only have few minor remarks. I wonder if the transfection of COCs would allow the miRNA mimics and inhibitors to reach the oocyte. A major part of the effects you see later may be due direct effects on the cumulus cells and other indirect effects mediated through the affected cumulus cells on oocytes. Please comment on that and consider to include that as a limitation in your discussion.
Line 139: please indicate if the n=240 and n=120 are per replicate or in total.
Line 156: Please expand on the target gene prediction procedure. Which groups have been compared to perform this prediction and to rule-out any non-specific effects of transfection or RNA sequences.
Author Response
We would like to thank the reviewer for her/his enthusiasm for the study and the constructive comments which have improved the manuscript. Please see the attachment including our response to each point raised.

Reviewer 2 Report
The article „Inhibition of miR-152 during in vitro maturation enhances the developmental potential of porcine embryos” submitted to the animals journal is focused on a very interesting subject of miRNA role during oocyte maturation and its impact on further embryo. The experiment is well designed and gives a lot of interesting data. However in my opinion there are some elements missing. E.g. the authors did not provide any data on cumulus cells, which would be very interesting in the whole environment-cumulus cells-oocyte pathway story. In the introduction it has been written that miR-152 affects the cell proliferation. It has also been described, how the oocyte and cumulus cells support each other in the development (secreted important factors). Therefore it could be suspected that the miRNA pathway of affecting the oocyte goes through the cumulus cells. I strongly recommend to include at least some gene expression analysis. It would show, whether the applied experimental systems affect cumulus cells, which ultimately support or inhibit oocyte developmental competence.
Other part of the story, that in my opinion is missing, is whether miR-152 is present in the follicular fluid? As far as I could read in the literature, some miRNAs are produced by the oocyte but quite a lot of them are transferred to the oocyte from the follicular environment. Do you have any data on this precise miRNA in porcine follicular fluid? It would be very important to know, especially that it is common to use follicular fluid as a supplement during IVM of porcine oocytes. Did you also use it?
There are also other minor comments to this manuscript:
- There were two types of samples analyzed with regard to gene expression – oocytes and embryos. In many fragments of the manuscript (summary, discussion) it is difficult to understand, which results are described, it is not precisely defined. Please correct this within the whole manuscript.
- In my opinion it is difficult to find the precise definition of the experimental groups. I know it is in the section 2.2, however it should be better defined. Besides the experimental group definition “Mimic” - I believe it does not really show the real definition of this group, which as far as I understood was overexpression of miR-152. Maybe it would be worth to change the names of the experimental groups?
Taking the above arguments under consideration I suggest the major revision of the manuscript.
Author Response

(The authors gave the same response as above.)

Reviewer 3 Report
Gad et el. present here a study on the role of miR-152, a highly conserved microRNA being mainly known to be involved in cell differentiation and proliferation. In a previous survey the authors studied the expression of miR-152 in growing oocytes.
Here they present novel data on the role of miR-152 in porcine oocyte and embryo developmental competence. The authors managed to manipulate the expression level of miR-152 during in vitro oocyte maturation. Maturation and Cleavage rate was comparably high in all groups (impressive numbers!). But down-regulation of miR-152 resulted in a significantly higher blastocyst rate compared to the up-regulated or control groups. There is a strong indication that miR-152 is targeting the IGF system. When artificially up-regulated, some quality-related genes in blastocysts were dysregulated.
The submission lies in the scope of the journal and all chapters are clearly and well written. I suggest to publish the paper in the journal.
Specific comments:
Could you please insert a sentence to discussion or introduction about what is known of miR-152 expression levels in in vivo oocytes before and after maturation? Clarifying of what is physiologic could contribute to the improvement of culture media. It might also be that in vitro culture has a high impact on expression levels even on your controls.
As far as I can see, incubation with transfection solution only occurred during the 2 days of maturation. Interestingly, the respective impact is still visible in the blastocysts. Since there is no difference in the cleavage rate, I would be curious at which embryo stage the development starts to differ. Is it just in the blastocysts? Could you provide data maybe to morula stage also? It would also contribute to the clarification of the event more precisely.
Could you please explain shortly in your manuscript why you decided to perform parthenogenetic activation instead of IVF or ICSI? Probably both methods do not work well in porcine. However, the embryonic development between real and parthenogenetic embryos differs in some species.
Line 132: I could not find an explanation for PZM 3 medium
Line 133: I would prefer “putative” instead of “tentative”
References: Line 375, 426,438,470: differences in format compared to the other references
Author Response

(The authors gave the same response as above.)

Round 2
Reviewer 2 Report
The authors concerned most of my doubts from the review. Although I still believe that including data on CC and FF would increase the value of this manuscript, I find the present form worth publishing.